# Fibrous or Prismatic? A Comparison of the Lamello-Fibrillar Nacre in Early Cambrian and Modern Lophotrochozoans

**DOI:** 10.3390/biology12010113

**Published:** 2023-01-11

**Authors:** Luoyang Li, Marissa J. Betts, Hao Yun, Bing Pan, Timothy P. Topper, Guoxiang Li, Xingliang Zhang, Christian B. Skovsted

**Affiliations:** 1Frontiers Science Center for Deep Ocean Multispheres and Earth System, Key Lab of Submarine Geosciences and Prospecting Techniques, Minister of Education and College of Marine Geosciences, Ocean University of China, Qingdao 266100, China; 2Laboratory for Marine Mineral Resources, National Laboratory for Marine Science and Technology (Qingdao), Qingdao 266237, China; 3Shaanxi Key Laboratory of Early Life and Environments, State Key Laboratory of Continental Dynamics and Department of Geology, Northwest University, Xi’an 710069, China; 4Palaeoscience Research Centre, School of Environmental and Rural Science, University of New England, Armidale, NSW 2351, Australia; 5State Key Laboratory of Palaeobiology and Stratigraphy, Nanjing Institute of Geology and Palaeontology, Chinese Academy of Sciences, Nanjing 210008, China; 6Department of Palaeobiology, Swedish Museum of Natural History, P.O. Box 50007, SE-104 05 Stockholm, Sweden

**Keywords:** Cambrian explosion, biomineralization, lamello-fibrillar nacre, mollusk, hyolith, annelid, seawater chemistry, Mongolia

## Abstract

**Simple Summary:**

The current understanding of the origin and rapid diversification of biomineralized skeletal animals and their environmental interactions during the Cambrian Radiation is largely dependent on the phosphatized skeletal remains of early animals. However, many questions remain unresolved about this early Cambrian animal skeletonization bio-event, as identification of primary mineralogy and microstructural organization of these earliest skeletons is often difficult. The comparison of previously poorly understood fibrous microstructures (represented in this study by lamello-fibrillar nacre in early Cambrian mollusk and hyolith shells) with their modern counterparts (coleoid cuttlebones and serpulid tubes) shows key differences in shell microstructures of these animal groups. For example, the aragonitic lamello-fibrillar nacre in some Cambrian shells is herein demonstrated to represent a primitive calcitic prismatic microstructure. This also demonstrates the prevalence of calcitic shells in the Terreneuvian, a period of time when seawater chemistry was thought to predominantly facilitate the precipitation of aragonite skeletons, suggesting that biomineralization of prismatic calcite is strongly biologically controlled.

**Abstract:**

The Precambrian–Cambrian interval saw the first appearance of disparate modern metazoan phyla equipped with a wide array of mineralized exo- and endo-skeletons. However, the current knowledge of this remarkable metazoan skeletonization bio-event and its environmental interactions is limited because uncertainties have persisted in determining the mineralogy, microstructure, and hierarchical complexity of these earliest animal skeletons. This study characterizes in detail a previously poorly understood fibrous microstructure—the lamello-fibrillar (LF) nacre—in early Cambrian mollusk and hyolith shells and compares it with shell microstructures in modern counterparts (coleoid cuttlebones and serpulid tubes). This comparative study highlights key differences in the LF nacre amongst different lophotrochozoan groups in terms of mineralogical compositions and architectural organization of crystals. The results demonstrate that the LF nacre is a microstructural motif confined to the Mollusca. This study demonstrates that similar fibrous microstructure in Cambrian mollusks and hyoliths actually represent a primitive type of prismatic microstructure constituted of calcitic prisms. Revision of these fibrous microstructures in Cambrian fossils demonstrates that calcitic shells are prevalent in the so-called aragonite sea of the earliest Cambrian. This has important implications for understanding the relationship between seawater chemistry and skeletal mineralogy at the time when skeletons were first acquired by early lophotrochozoan biomineralizers.

## 1. Introduction

The longer a skeletal fossil is buried, the less information regarding biomineralization it tends to retain after lithification and diagenetic alteration [1,2,3]. Fossil preservation is inevitably biased and involves complex diagenetic processes such as compaction, replacement, and recrystallization, which cause great difficulties when investigating the biomineralization of fossilized hard tissues. This is especially the case with the diverse phosphatized skeletal remains of early animals, called small shelly fossils (SSF), extracted from the latest Ediacaran to Cambrian phosphorites, at the time when metazoans first emerged and acquired the ability to precipitate different biomaterials [4]. In recent years, growing interests and intensive biomineralization surveys on SSF have provided direct evidence for deciphering the onset and diversification of skeletal biomineralization during the explosive radiation of metazoans across the Precambrian–Cambrian transition [5,6,7]. Biomineralization has also proven to be a very useful character in phylogenetic analyses, especially for resolving the biological affinities of a variety of enigmatic Cambrian fossil groups, e.g., hyoliths [8,9], tommotiids [10], and chancelloriids [11]. Despite these advances and although it is possible for the microstructural details of many skeletal fossils to be preserved by secondary phosphate minerals (e.g., phosphatization of mollusk [12] and brachiopod shells [13]), uncertainties have persisted in determining primary mineralogies and architectural organizations of these earliest animal skeletons.

The synthesis of mineralized hard tissues by organisms is regulated by the orchestration of biological (self-assembled organic matrix) and/or physiochemical processes (crystal growth competition). The cooperation between the two processes, i.e., the extent of biological control over skeletal biomineralization, varies among organisms. Accordingly, skeleton-secreting mechanisms are often categorized into biologically induced (BIM) versus biologically controlled (BCM) mineralizing types [14]. Throughout the Phanerozoic, the large-scale secular variation of calcite-aragonite polymorphs in abiogenic carbonate deposits (e.g., ooids and marine carbonate cements) and simple reef- and sediment-forming marine hypercalcifying organisms (e.g., corals and algae) appear to have been profoundly influenced by seawater chemical perturbations, especially the Magnesium/Calcium molar ratio (mMg/Ca ratio) [15,16]. High-Mg calcite and aragonite were favored in the so-called aragonite sea (mMg/Ca ratio > 2), while the calcite sea (mMg/Ca ratio < 2) is associated with the precipitation of low-Mg calcite skeletons. However, long-standing debate exists with respect to whether and to what extent did changing seawater chemistry influence carbonate polymorphs in sophisticated BCM skeletons, particularly at the time when metazoans first acquired the ability to biomineralize skeletons [4,17,18,19]. Laboratory experiments have been conducted to decipher the effects of seawater chemistry on skeletal mineralogy in modern taxa, e.g., [20,21]. However, many questions fundamental to unraveling the Cambrian metazoan skeletonization bio-event remain unresolved, primarily because the “basic data” regarding biomineralization of very early animal skeletons (e.g., primary mineralogy and skeletal microstructures) are difficult to obtain.

To better understand how the earliest calcifying animals regulated the production of calcareous shells and the relationship between skeletal mineralogy and seawater chemistry, we present a close examination of previously poorly understood fibrous shell microstructures of early Cambrian SSF and compare them with their modern counterparts. Special attention is paid to the lamello-fibrillar (LF) nacre, a representative fibrous microstructure commonly reported from early Cambrian mollusks and hyoliths (e.g., [22]). The LF nacre is also a unique microstructural form of some cephalopods (cuttlebones and belemnitids), and probably some serpulid annelids.

While extant serpulids, coleoids, and Cambrian mollusks and hyoliths are able to secrete calcareous skeletons with LF nacre or LF nacre-like fibrous microstructures (see Figure 1), significant differences in their mineralogy, crystal shape, size, and hierarchical organizations have often been overlooked. Such differences are controlled by biomineralization mechanisms particular to each group (see below). Because the claim of the existence of LF nacre in Cambrian mollusk and hyolith shells rests largely on the observation of fibrous morphology and a crisscrossed texture on the surface of internal mold specimens, a whole picture of the microstructure in three-dimensionality is rarely known. Our comparative study provides crucial new information in characterizing the LF nacre and similar microstructures in modern coleoids and serpulids. Some, but not all, previously described fibrous microstructures of Cambrian mollusks are confidentially re-interpreted to be a primitive type of calcitic prismatic microstructure rather than a nacreous LF form as in coleoid mollusks. Despite superficial similarity in their fibrous and crisscrossed appearance, significant differences are highlighted amongst various lophotrochozoan groups in terms of their microstructure and mineralogy.

## 2. Fiber and Lamello-Fibrillar Nacre in Lophotrochozoans

Fibers, a morphological term broadly referring to a wide array of needle, rod, and elongate lath-shaped crystals, are one of the principal structural elements in biogenetic materials. They share a generally fibrous morphology but exhibit high variability in crystal shape and size, crystallographic orientation, and hierarchical organization among different biocalcifying systems and their morphological features are usually not closely related to mineralogical compositions. Common fibrous microstructures are the calcitic fibrous prismatic form of mytilid bivalve shells [23], the primary and fibrous shell layer of calcitic brachiopods [24,25], aragonitic fibers of corals [26], and even the calcite-reinforced chitin fibers of arthropod cuticles [27].

The LF nacre is a fibrous microstructure unique to biocalcifying lophotrochozoans. It is an aragonitic, laminar, plywood-like shell microstructure consisting of sheets (first-order unit) of more or less mutually parallel, horizontal crystals (second-order unit), with crystal orientations differing in successive laminae or sheets, which result in a characteristic fibrous and crisscrossed pattern on the depositional surface of shells ([28], also refer to Figure 1D for schematic illustrations). This laminar microstructure, also called “type-II nacre”, was initially described in extant coleoid cephalopods [29], though it lacks the typical stacked towers of aragonitic hexagonal tablets as in the columnar nacre of cephalopods. The LF nacre occurs commonly in septa and septal necks of cuttlebones, belemnitids, and spirulids (Figure 1C) [30,31] (see review in [32]). The possible presence of LF nacre was also reported in the recent vetigastropod *Stomatella planulata* [33] and the Eocene gastropod *Clavilithes* [34].

The ordered-chevron (OC) microstructure of some calcareous serpulid tubes (annelids) is reminiscent of the LF nacre of coleoids in that it exhibits a similar crisscrossed character on the depositional surface of the tube. The OC microstructure is the most hierarchically complex architecture in serpulid tubes and, while its formation is organically mediated, it is achieved under much weaker biocalcification controls compared to that of mollusks [35,36]. The OC microstructure was often compared to, and actually described as, the LF nacre of mollusks by some authors, e.g., [37,38,39]. However, a recent study revealed clear differences between the OC microstructure of serpulid tubes and the “true” LF nacre of coleoid mollusks with respect to the mineralogy and hierarchical organization of crystals [34]. According to Buckman et al. [34], the arrangements of calcitic prisms of the OC microstructure are more irregular and the long axis of calcitic prisms may incline at a degree of up to 45° with respect to the plane of the tube’s growth lamellae, which is different from the microstructural organization of acicular aragonitic crystals in the LF nacre of mollusks.

Palaeontological surveys on Cambrian molluskan biomineralization reveal a LF nacre-like fibrous microstructure that consists of fine fibrous crystals arranged in stacked laminae with a crisscrossed pattern (Figure 1A). The term “lamello-fibrillar” was used for the first time to describe fibrous forms with a “stepwise” microstructure (named as the foliated aragonite in [40]) in the Cambrian bivalve *Fordilla*/*Pojetaia* and proto-rostroconch *Anabarella*/*Watsonella* [41]. Subsequently, “lamello-fibrillar” was used to describe similar fibrous microstructure in the scaly shells of early Cambrian maikhanellids and several cap-shaped and coiled helcionellidan mollusks such as *Ilsanella* and *Oelandiella* [42,43]. It was further proposed that the LF nacre might represent a basic and primitive microstructural type of the inner shell layer of early Cambrian mollusks [43]. Meanwhile, the outer shell layer of early mollusks was thought to have mainly consisted of an aragonitic prismatic microstructure, the presence of which was inferred based on the preserved polygonal texture on the surface of internal molds [41]. This mode was widely accepted and similar fibrous microstructures often described as the LF nacre were reported in various lineages of early mollusks, including shells of possible stem-group polyplacophoran *Ocruranus* [44] and stem-group gastropods *Aldanella* and *Pelagiella* [5,12].

Additionally, the possible existence of LF nacre has been reported in some Cambrian hyoliths. Hyoliths were a group of problematic lophotrochozoans in Paleozoic marine communities, divided into two main subgroups: hyolithids and orthothecids [45]. They possessed a calcareous conch and a lid-like operculum and an additional pair of spines called “helens” in the hyolithids [46,47]. Several recent studies focusing on hyolith soft tissues preserved in Cambrian Konservat-Lagerstätten have suggested that they may have had a tentaculate filter-feeding organ similar to the lophophore of lophophorates, a supergroup encompassing brachiopods, bryozoans, and phoronids [48]. However, in the perspective of shell microstructures, the striking skeletal similarities between hyoliths and mollusks indicate a much closer phylogenetic link between these two lophotrochozoan groups [7,8,9]. A variety of fibrous microstructures have been described in several early Cambrian hyolith taxa, such as the “fiber bundles” of *Majatheca* [49] and more simple parallel fibers of *Paramicrocornus* [50]. Of these, fibrous crystallites of the inner shell layer of *Cupitheca* and *Protowenella* show an interesting crisscrossed organization that are considered to be equivalent to the LF nacre of coleoid mollusks [51]. Fibrous microstructures interpreted as LF nacre were also documented in both hyolithid and orthothecid conchs from the middle Cambrian (Drumian, Floran) Gowers Formation of the eastern Georgina Basin, Queensland, Australia [22]. We also note that while the recent finding of complete specimens of *Protowenella* from Greenland suggested a hyolith affinity (*Protowenella* has long been regarded as stem-group gastropod [52,53]), hyoliths and mollusks are phylogenetically closely-related and deeply homologous in shell biomineralization [8]. Another recent significant discovery of heavily-coiled *Cupitheca* shells from Siberia [54] suggests that morphologically, the definitions of these two groups are becoming blurred.

## 3. Geological Settings, Materials and Methods

The lower Cambrian carbonate-siliciclastic sequences in the Zavkhan Basin, Southwestern Mongolia host a treasure trove of SSFs, which have received extensive attention from paleontologists and geologists for over half a century. See [55] for a comprehensive review of the chemostratigraphy, biostratigraphy, and lithostratigraphy of the Zuun Arts–Bayangol succession. The limestone samples for the present study come from the Bayangol Formation; horizon BAY2/596 in the BG2 section and 341.0 m true thickness from the base of the section (base of section GPS: 46°42′11.0″ N, 96°18′44.5″ E). The samples were collected during the 2018 fieldtrip season led by C.B.S (Figure 2).

The limestone samples were broken and macerated in diluted 5% acetic acid solution in the laboratory. The acid-resistant residues were washed, wet-sieved carefully, and air-dried at room temperature. Then, the mollusks together with other skeletal fossils were manually picked under stereomicroscopy. The specimens were mounted on stubs, sputter-coated with gold and examined with a FEI Quanta FEG 650 Scanning Electron Microscopy (SEM) at an accelerating voltage of 15 KV at the Swedish Museum of Natural History. All the imaged Mongolian material is deposited at the Swedish Museum of Natural History (NRM).

The extant cuttlebones and serpulid tubes were obtained commercially from the Manyuwan seafood market in Qingdao City, which was originally collected from the Yellow Sea of China, identified as *Sepia esculenta* Hoyle, 1885, and *Hydroides elegans* Haswell, 1883, respectively. The specimens were physically broken into small fragments then mounted on stubs, sputter-coated with gold, and examined with a FEI Quanta FEG 450 SEM at an accelerating voltage of 15 KV at the Shaanxi Key Laboratory of Early Life and Environments, Northwest University. All the imaged extant material is deposited at the College of Marine Geoscience, Ocean University of China (OUCCMG).

## 4. Results

### 4.1. Lamello-Fibrillar Nacre of Extant Coleoid Cuttlebones

Our observations of the LF nacre in *Sepia esculenta* are in accordance with the recent comprehensive investigation of the microstructure in the modern cuttlefish *Sepia officinalis* [32]. Transverse sectioning of *S. esculenta* cuttlebone reveals a dorsal shield and a ventral chambered zone that consists of septa and pillars (Figure 3A). Each septum is composed of three superimposed structural layers: outer and inner layers with prismatic organization sandwiching a thicker LF nacre layer (Figure 3B). The LF nacre is composed of a series of horizontal lamellae or sheets to the depositional surface of the septum. The fibers are oriented in multiple directions on different lamellae, forming a characteristic crisscrossed pattern on the internal surface of the septum (Figure 3E,F). The individual fibers are needle-like, 150–200 nm wide (N = 10), 1–10 μm long (N = 10), contain high organic contents, and have a nanogranular component (Figure 3H,I).

### 4.2. Ordered-Chevron Microstructure of Serpulid Tubes

The observations on the fractured surface of the modern serpulid *Hydroides elegans* tube (Figure 4A) reveal a distinct two-layered structural organization. The thin outer layer has a prismatic microstructure, while the thick inner layer has an ordered-chevron microstructure in which each growth lamellae is approximately 10 μm in maximum thickness (Figure 4B,C). In the inner layer, characteristic rod-shaped and prismatic crystallites are mutually parallel and horizontally oriented within the lamella, but their orientations are variable in successive growth increments (Figure 4D,E). The arrangement of crystals as such results in a crisscrossed texture on the internal surface of the tube (Figure 4F,G). The alignment of elongated, rod-shaped crystals follows the directions of pre-existing organic sheaths (Figure 4H). Close-up examination of the mineral prisms shows typical rhombohedral morphology indicative of calcite. The individual prisms are single crystals of calcite, with a maximum of 1 μm in diameter and a maximum of 10 μm in length, which are composed of dense nanogranules (Figure 4I).

### 4.3. Prismatic Microstructure of Cambrian Postacanthella Shells

Delicate preservation of the inter- and intra-prismatic organic matrix indicates a calcitic prismatic microstructure of the shell [56]. The new specimens of *Postacanthella* (herein) have prismatic microstructure that can be divided into two sub-types based on key differences in crystal orientations and arrangements. These are regularly oriented simple prismatic microstructure (ROSP) and irregularly oriented crisscrossed prismatic microstructure (IOCP).

The columnar prisms of the ROSP microstructure appear to radiate from the presumed apex of the shell (Figure 5A,B). The individual small prisms (~1 μm in diameter) consist of mutually paralleled sub-structural units coated by a thin organic sheath. By contrast, large mature prisms (up to 10 μm in diameter) are irregularly divided by intraprismatic organic membranes (Figure 5D,E). The columnar prisms of the ROSP layer are mutually paralleled and are inclined at an acute angle or are perpendicular to the internal surface of the shell.

The IOCP microstructure is characterized by irregularly oriented calcitic prisms. Columnar prisms of the IOCP microstructure have a variety of orientations; they may be angled at high degrees to, perpendicular to (Figure 6A,B), or parallel to (Figure 6C) the depositional surface of the shell. Similar to the ROSP microstructure, the original mineralogy of the prisms has been altered but the associated organic envelope surrounding the mineral phases have been replicated exquisitely by phosphate (Figure 6D,E). Most intriguingly, the individual prisms tend to be horizontal close to the internal-most surface of shell and intersect with the adjacent prisms. Hence, they exhibit an unusual crisscrossed morphology, reminiscent of the LF nacre of coleoids and the ordered-chevron microstructure of the serpulid tubes (Figure 6F).

## 5. Discussion

### 5.1. Preservation and Reconstruction of Postacanthella Shells

The *Postacanthella* specimens from the Bayangol Formation are mostly preserved as phosphatic internal molds, which is consistent with the preservation of other molluskan taxa from the formation and elsewhere, e.g., [57,58]. As such, the original calcareous shells have generally been altered or etched away via diagenesis and laboratory acid processing. However, close-up examinations of the steinkerns show that their surfaces frequently retain remnants of the shell-secreting organic matrix that has been preserved in high fidelity by phosphate. In these cases, the prismatic organic envelope that primarily encased a mineral prism has left a delicate polygonal and cell-like texture. The original tubules or pores of the shell were secondarily infilled by phosphatic mineral aggregates, forming a tower-like feature or small tubercles on the surface of the specimens. The tower-like phosphatic infillings or tubercles are arranged regularly in commarginal rows and growth increments (lines) are often visible (Figure 7E).

Some specimens preserve a distinct commarginal line or scar along the aperture of the shell. This commarginal line is here interpreted to represent the mantle edge of the animal (Figure 7B). In some other specimens, a characteristic U-shaped muscle attachment zone corresponding to the myostracum is discernable (Figure 7C). The preservation of the mantle line and muscle attachment scar never co-occur within one single specimen of our collections. The mantle margin of *Postacanthella*—assumed to be retractable inwards and extendable beyond the shell margin when the animal was alive—always occurs closer to the edge of the specimen relative to the muscle attachment zone. Their spatial distribution on internal molds appears to be identical to that in modern limpet-shaped monoplacophoran and gastropod shells. The ROSP microstructure, forming the outermost shell layer (M + 2), occurs adjacent to the shell aperture in comparison with the IOCP type that is considered to constitute the inner layer (M + 1) of the shell.

In modern shells, one or two additional microstructural layers (M − 1, M − 2) occur that are separated from external shell layers by the myostracum. This might be a similar case for *Postacanthella*, in that its calcareous shell is composed of multiple layers as proposed here (Figure 7A). Our observations demonstrate the existence of at least two prismatic structural layers (M + 2 and M + 1) in *Postacanthella* shells, but the structure of a myostracum (the thin layer composed of aragonitic prisms in modern mollusk shells) and secretion of a M − 1 layer are still not clear. In addition, no clear discontinuities or interface have been observed between the outer ROSP and inner IOCP layers. Instead, the transition between these two prismatic microstructures seems to be progressive. The orientation of calcitic prisms in the outer ROSP microstructure is constantly perpendicular to or highly inclined to the shell surface but gradually transitions into a more irregularly oriented pattern within the inner IOCP microstructure. The calcitic prisms are continuously secreted throughout the thickness of the shell. Furthermore, no derived laminar microstructural types such as the foliated aragonite of some molluskan taxa are seen in *Postacanthella*.

### 5.2. Lamello-Fibrillar or Prismatic?

This comparative study shows that while the LF nacre of coleoid cuttlebones, the OC microstructure of serpulid tubes, and the prismatic microstructure (especially the IOCP type) of Cambrian helcionelloid shells are superficially similar in terms of their fibrous morphology and crisscrossed organization of crystallites, significant differences exist with respect to their crystal shape, size, orientation, and mineralogy. The differences between the LF nacre (coleoids) and the OC microstructure (serpulids) are clear: coleoids secrete characteristic elongated, needle-shaped crystallites of aragonite parallel to the shell depositional surface, while serpulids typically precipitate rhombohedral crystals of calcite that are often inclined to the growth lamellae of tubes. The LF nacre is better considered to be a microstructural motif confined to the Mollusca [34].

There is a tendency for mineral prisms in the IOCP microstructure to become gradually horizontally oriented with respect to the internal-most surface of the shell. Thus, via competitive crystal growth, individual prisms often intersect and overlap with adjacent prisms, thereby exhibiting a crisscrossed pattern on the internal surface of the shell, reminiscent of LF nacre. However, the IOCP microstructure can be easily distinguished from LF nacre based on its characteristic irregularly oriented prisms and calcitic composition. There are also intriguing similarities between the IOCP microstructure and the irregularly oriented prismatic (IOP) microstructure of some serpulid tubes, in which calcitic prisms also show high irregularity in multiple directions, with individual prisms internally consisting of elongate, mutually parallel second-ordered subunits [36,37]. However, in *Postacanthella*, the IOCP form is clearly organically enriched with the basic prismatic building blocks encased within an inter- and intra-prismatic organic matrix scaffold. This indicates that the assembly of organic-encased calcitic prisms in the IOCP microstructure of *Postacanthella* is under strong biological control. In contrast, the calcitic prisms of the OC microstructure develop a typical rhombohedral shape, indicating limited influence on crystal growth exerted by the associated organic matrix sheaths in serpulid tubes.

Because the microstructural details of Cambrian mollusks are mainly preserved as imprints or phosphatic replicas on the surface of internal molds, the IOCP microstructure is likely to be misidentified as the LF nacre based on its crisscrossed organization of crystallites and fibrous morphology. Fibrous microstructures previously referred to as the LF nacre in Cambrian maikhanellids *Ramenta* and helcionellids *Ilsanella* and *Oelandiella* are confidentially re-interpreted to be IOCP microstructure constituted of calcitic prisms. For example, the specimen in Figure 6C herein appears to be identical to the figured fibrous microstructure of *Ramen tacambrina* (plate 4, Figure 10 in [42]), in which the “lamello-fibrillar” microstructure was initially identified in Cambrian molluskan shells. Re-evaluation of those fibrous microstructures of Cambrian mollusks is necessary but is currently heavily limited by fossil preservation in previous publications. Nevertheless, the shell microstructure of *Pelagiella subangulata* has fibrous, acicular crystals consistently aligned in a radial direction and is therefore considered to be aragonitic [59]. These horizontal fibers are different from the IOCP microstructure of *Postacanthella* shells consisting of calcitic prisms. Moreover, the fibrous microstructure with a comparable crisscrossed pattern in hyoliths may also be an IOCP microstructure, further adding to the striking skeletal similarities between mollusks and hyoliths.

### 5.3. Biomineralizing a Prismatic Shell in the Terreneuvian Aragonite Sea

Prismatic microstructures are commonly formed in the outermost layer and muscle attachment layer (myostracum) of a shell. The secretion of mineral prisms is initiated on the inner surface of the protective outer organic layer (the periostracum) and growth is guided by the incorporated prismatic organic matrix [60,61,62]. The four main types are well-characterized in terms of the hierarchical organization and mineralogy of shell microstructure. These are the simple prismatic, fibrous prismatic, spherulitic prismatic, and compound prismatic forms. Each type further possesses several different variations, for example, the simple prismatic can be divided into regular/irregular simple prismatic, pavement prismatic, or radial elongate prismatic, showing the extremely high diversity of prismatic organizations evolved over hundreds of millions of years [28].

The IOCP form is apparently unique among molluskan prismatic forms, but to the best of our knowledge, such highly irregular organization of prisms has never been reported in any modern or fossil mollusk shells. Instead, the IOCP form bears more similarities with the calcitic IOP microstructure of some serpulid tubes [36,37]. This microstructural similarity is in accordance with other morphological features shared between stem-group members of these two sister clades, e.g., chaetae [63]. However, these groups are distinguishable based on biomineralization mechanisms and the overall hierarchical organization and complexity of shell/tube microstructures (mentioned above).

Columnar prisms in some early Cambrian mollusks were thought to have a spherulitic organization consisting of aragonite [64]. However, this contrasts with the simple prismatic shells composed of calcite in *Postacanthella*. Our close examination of fibrous microstructures in early Cambrian mollusks (including in hyoliths) suggests that calcitic shells were not uncommon in the Terreneuvian, during an aragonite sea when aragonite shells were presumably favored [65]. Prismatic microstructure—likely a primitive character—is the dominant microstructure in the earliest Cambrian mollusks that biomineralized shells, while laminar microstructure such as foliated aragonite and crossed-lamellar forms tend to occur in more derived members of helcionelloids and gastropods. Throughout the rapid speciation of mollusks in the early Cambrian, many taxa inherited a prismatic shell from their latest mineralized common ancestors, indicating a deep homology at the onset of prismatic biomineralization.

The processes determining the acquisition of particular shell compositions in mollusk groups are complex. Despite the homologous nature of prismatic shell microstructure in mollusks, the acquisition of particular calcite/aragonite polymorphs might have been species-specific, reflecting a variety of responses to surrounding seawater environments. The mineralogical evolution of mollusks also appears to have followed the transition of aragonite to calcite seas during the early Cambrian, indicating that changing seawater chemistry exerted some influence on shell mineralogy when new species first emerged [12,66]. Yet, the adoption of calcite in *Postacanthella* and others is evidence that this process can also be strongly biologically controlled, despite the production of a primitive prismatic shell with high irregularities in crystal organization seeming to be less well-controlled by the associated prismatic organic matrix compared to that of the highly organized prismatic forms of modern mollusks.

It is currently difficult to precisely weigh the effects of biological and/or environmental physiochemical factors on the shell mineralogy of Cambrian stem-group mollusks. However, the obvious increase in shell thickness and toughness from the early Cambrian onwards implies increasingly enhanced biological controls over shell calcification, likely in response to increasing predatory pressure (e.g., during the Great Ordovician Biodiversification Event). Over time, the shell mineralogy appears to have become evolutionarily conserved regardless of seawater chemical perturbations and prismatic organization has become one of the most successful shell microstructures for protecting the vulnerable soft morphology of mollusks.

## 6. Conclusions

This study compared the lamello-fibrillar nacre and similar fibrous microstructures in Cambrian and modern lophotrochozoans and demonstrated that the lamello-fibrillar nacre is a microstructural motif confined to mollusks, that differs largely from the ordered-chevron form in serpulid tubes. Similar fibrous microstructures previously described in many early Cambrian mollusk and hyolith shells were re-interpreted to be a primitive prismatic microstructure rather than the nacreous LF form (as in coleoids). The helcionelloid mollusk *Postacanthella* from the lowermost Cambrian (Terreneuvian) of southwestern Mongolia possessed a calcitic shell with several superimposed structural layers including regularly oriented simple prismatic microstructure and irregularly oriented crisscrossed prismatic microstructure. *Postacanthella* is evidence that calcitic mollusk shells co-occurred with aragonitic shells during the aragonite sea of the earliest Cambrian. This suggests that molluskan taxa exhibited a variety of responses to fluctuating seawater environments during the early Cambrian Radiation and highlights the importance of biologically controlled biomineralization during the most nascent phases of shell evolution.

## Figures and Tables

**Figure 1 biology-12-00113-f001:**
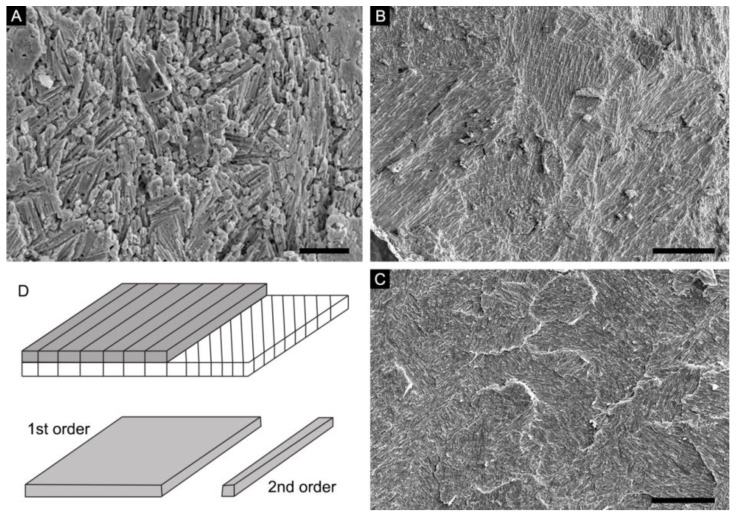
Schematic diagram of the LF nacre and similar fibrous microstructure in extant and fossil groups (**A**) Specimen NRM Mo196854, prismatic microstructure showing a fibrous and crisscrossed appearance on surface of internal mold of a Cambrian helcionelloid *Postacanthella* specimen. (**B**) Specimen OUCCMG An00001, similar fibrous and crisscrossed morphology of the ordered-chevron microstructure seen on the internal surface of modern serpulid tube. (**C**) Specimen OUCCMG Mo00001, overview morphology of the LF nacre of extant cuttlebones. (**D**) Schematic diagram of the LF nacre showing hierarchical orderings. Scale bars: 20 μm in (**A**–**C**).

**Figure 2 biology-12-00113-f002:**
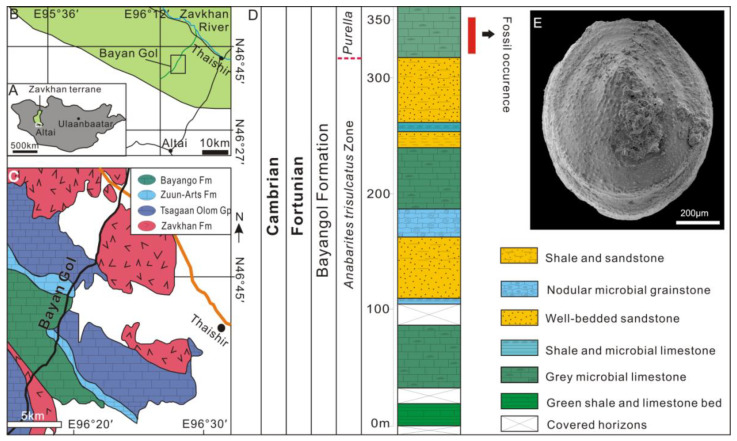
Geological setting, lithostratigraphy, and biostratigraphy. (**A**) Locality map of the Zavkhan terrane, southwestern Mongolia. (**B**) Location of Bayan Gol in the Zavkhan Basin. (**C**) Geological map of Bayan Gol and the location of section BAY2, GPS: 46°42′11.0″ N/96°18′44.5″ E. (**D**) Lithostratigraphic columns of the Bayangol Formation in the BAY2 section and fossil occurrence horizon. The boundary of the small shelly fossil *Anabarites trisulcatus* Zone and *Purella* Zone is temporarily placed at 335 m above the base of the section. (**E**) Specimen NRM Mo196855, overall morphology of cap-shaped helcionelloid *Postacanthella voronini*.

**Figure 3 biology-12-00113-f003:**
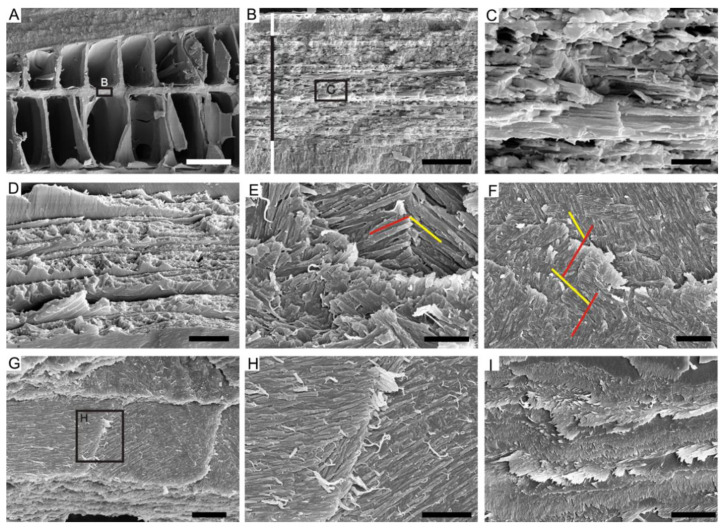
The LF nacre of extant coleoid *Sepia esculenta*. (**A**) Specimen OUCCMG Mo00002, transverse section showing the dorsal shield and ventral chambered zone. (**B**) Specimen OUCCMG Mo00003, close-up examination of the septum showing three superimposed structural layers; prismatic form marked by white bars, LF nacre layer marked by black bar. (**C**) Amplification of (**B**) showing the stacked lamellae of the LF nacre. (**D**) Specimen OUCCMG Mo00004, fractural surface showing stacked lamellae of the LF nacre. (**E**,**F**) Specimen OUCCMG Mo00005, acicular aragonite crystals orient differently in different lamellae, marked by red and yellow lines, plane view. (**G**–**I**) Specimen OUCCMG Mo00006, acicular crystals incorporated with high organic contents. Scale bars: 100 μm (**A**), 10 μm (**I**), 5 μm (**B**,**F**,**G**), 2 μm (**D**,**H**), and 1 μm (**C**,**E**).

**Figure 4 biology-12-00113-f004:**
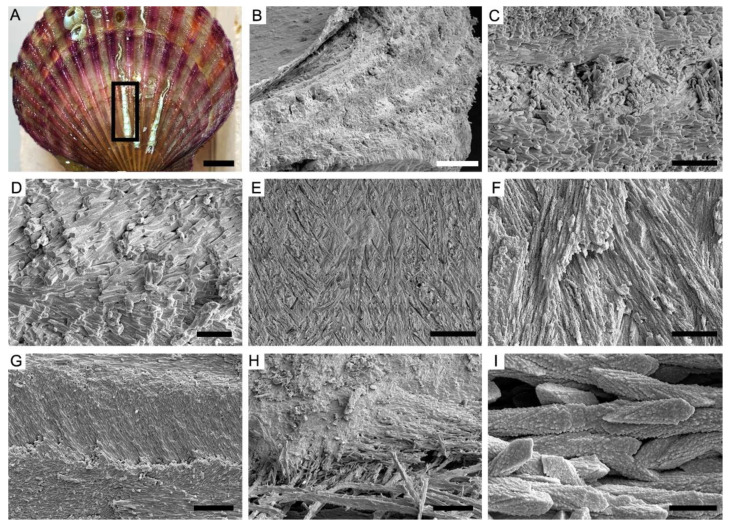
The ordered-chevron microstructure of serpulid tubes *Hydroides elegans*. (**A**) Specimen OUCCMG An00002, serpulid tubes encrusted on surface of bivalve *Azumapecten farreri* shells. (**B**–**D**) Specimen OUCCMG An00003, fractural surface of tube showing distinct growth increments of the ordered-chevron form. (**E**–**G**) Specimen OUCCMG An00004, fibrous and crisscrossed organization of crystals on internal surface of the tube; different orientations of crystals are marked by yellow and red lines, respectively. (**H**) Specimen OUCCMG An00005, close-up image showing differently oriented prismatic crystals in adjacent lamellae (marked by different color lines) and organic matrix sheath. (**I**) Individual prisms have a rhombohedral morphology. Scale bars: 1 cm (**A**), 50 μm (**E**), 20 μm (**B**,**H**), 10 μm (**G**), 5 μm (**C**,**D**,**F**), and 2 μm (**I**).

**Figure 5 biology-12-00113-f005:**
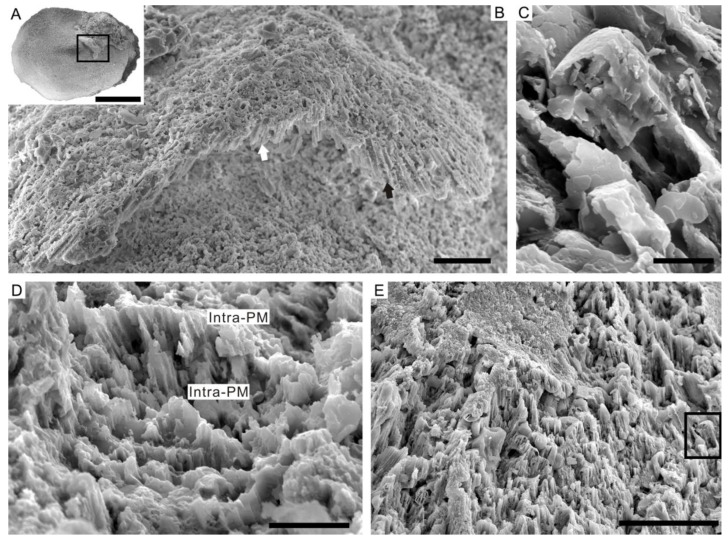
Regularly oriented simple prismatic microstructure of early Cambrian *Postacanthella* shells. (**A**) Specimen NRM Mo196856, overall morphology of cap-shaped *Postacanthella*, preserved as phosphatic internal mold. (**B**) Specimen NRM Mo196857, shell remnants showing the regularly oriented mineral prisms of the ROSP form. (**C**) Amplification of (**E**), showing individual prisms. (**D**) Specimen NRM Mo196858, phosphatized organic remnants interpreted as the inter- and intra-prismatic organic matrix. (**E**) Specimen NRM Mo196859, mineral prisms are highly inclined or perpendicular to shell surface. Scale bars: 500 μm (**A**), 20 μm (**B**,**E**), 5 μm (**D**), and 2 μm (**C**).

**Figure 6 biology-12-00113-f006:**
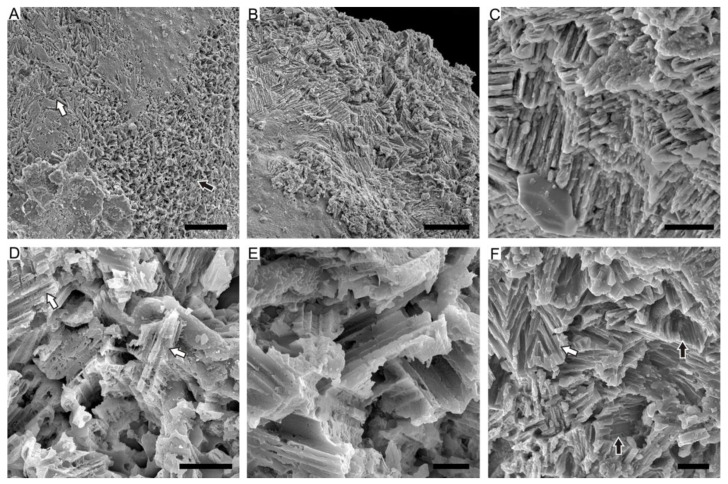
The irregularly oriented crisscrossed prismatic microstructure of early Cambrian *Postacanthella* shells. (**A**) Specimen NRM Mo196860, fibrous appearance and preservation of horizontally oriented (white arrow) and vertically oriented (black arrow) prisms on surface of phosphatic internal mold specimen. (**B**) Specimen NRM Mo196861, irregular orientation of mineral prisms. (**C**) Specimen NRM Mo196862, individual prisms intersect and overlap with adjacent prisms. (**D**,**E**) Specimen NRM Mo196867, irregular orientations of mineral prisms. (**F**) Specimen NRM Mo196863, individual prisms show second-order sub-structural units (black and white arrows). Scale bars: 50 μm (**A**), 20 μm (**B**), 5 μm (**C**,**D**), and 3 μm (**E**,**F**).

**Figure 7 biology-12-00113-f007:**
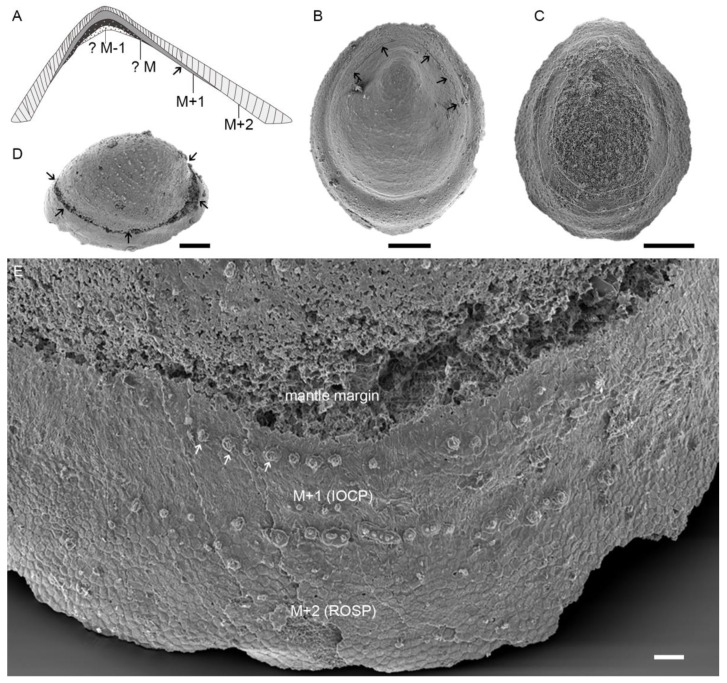
Preservation and reconstruction of early Cambrian *Postacanthella* shell. (**A**) Schematic diagram showing the reconstruction of *Postacanthella* shells, which is comparable with extant limpet-shaped shells. (**B**) Specimen NRM Mo196864, black arrows marking the commarginal line preserved on surface of internal mold. (**C**) Specimen NRM Mo196865, preservation of U-shaped muscle attachment zone. (**D**) Commarginal scar marked by black arrows. (**E**) Specimen NRM Mo196866, preservation of different shell layers with particular prismatic microstructures on surface of internal mold. Scale bars: 200 μm (**B**–**D**), 20 μm (**E**).

## Data Availability

Not applicable.

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
