# Peer review of "Fibrous or Prismatic? A Comparison of the Lamello-Fibrillar Nacre in Early Cambrian and Modern Lophotrochozoans"

_biology, 2023, doi:10.3390/biology12010113_

Round 1

Reviewer 1 Report

i highly appreciated to read this interesting manuscript. to me, it's a minor revision.

two points should be already mentioned here:

-it seems you did not clearly assign the specimens to specimen numbers. should be carefully checked.

- you mention only very briefly the alternative phylogenetic position, i.e. lophophorates rather than molluscs, published in [47]. with the serpulid worm, you include a lophotrochozoan, that's fine, but i would expect also a comparison with lophophorate skeletal microstructure to make your discussion consistent and comprehensive. there are papers, right?  

Author Response

it seems these specimen numbers are not correct. should the XX be replaced by a number?

Yes, but all specimen numbers shall be given when a paper is formally accepted for publication by the journal. Specimen numbers have been added in the revised version.

these taxon names are so annoying. can you please use the proper taxon names at least when you mention them for the first time?

OK, revised.

consisted of?

Right and revised.

you dont mention shell microstructure of lophophorates, e.g.brachiopds, as compared to the taxa studied here. i think this would be important to clarify previous and earlier homology assumptions

The lamello-fibrillar structure studied in current work is known only in mollusk, annelid and some cambrian skeletal fossils. This is the reason why we did not mention the biomineralization of lophophorate groups (brachiopods, phoronids and bryozoans).

i dont understand what "share deeply-homologous" means. in addition notions like more or less homologous or % homology are odd.

Revised.

didnt you introduce an abbreviation?

Yes, revised.

give full species names when a species is mentioned for the first time

Revised.

species? citation of earlier study? you cite one in the other results chapters....

information added.

indicate its a species of bivalvia

Revised.

again, specimen numbers are not clear. check throughout manuscript

Revised.

1 full stop is sufficient ;-)

Revised.

Reviewer 2 Report

This is a fine contribution to the understanding of evolution of exoskeletons and biomineralization. Only few suggestions to improve the prsentation are included in the attached file.

Author Response

is?

Right and revised.

perhaps you want to change position of D and C, so the typical figure arrangement is followed

I prefer to insert the cartoon diagram in the left panel of the plate, which, in my eyes, looks more beautiful.

Is this grey microbial limestone? the colour does not fit with any of the ones in reference colours for lithology

Yes, I suggest we could keep this in the current manuscript, which remains consistent with our previous publications.

Bayangol?

Yes, for the Formation

How is the correct name? Bayan Gol or Bayangol?

“Bayan Gol” is the name of the location, but for the formation, should be written as the "Bayangol".

Sorry if I was wrong, maybe the name of the region is different to that of the formation, but please check anyway.

We have checked it throughout the main text carefully.

Although obvious, please label the image as "A"

Added.

erase extra dot

Revised.